# Melatonin-Assisted Cisplatin Suppresses Urinary Bladder Cancer Cell Proliferation and Growth through Inhibiting PrP^C^-Regulated Cell Stress and Cell Proliferation Signaling

**DOI:** 10.3390/ijms24043353

**Published:** 2023-02-08

**Authors:** Chih-Chao Yang, Fei-Chi Chuang, Chia-Lo Chang, Chi-Ruei Huang, Hong-Hwa Chen, Hon-Kan Yip, Yen-Ta Chen

**Affiliations:** 1Division of Nephrology, Department of Internal Medicine, Kaohsiung Chang Gung Memorial Hospital, Chang Gung University College of Medicine, Kaohsiung 83301, Taiwan; 2Department of Obstetrics and Gynecology, Kaohsiung Chang Gung Memorial Hospital, Chang Gung University College of Medicine, Kaohsiung 83301, Taiwan; 3Division of Colorectal Surgery, Department of Surgery, Kaohsiung Chang Gung Memorial Hospital, Chang Gung University College of Medicine, Kaohsiung 83301, Taiwan; 4Division of Cardiology, Department of Internal Medicine, Kaohsiung Chang Gung Memorial Hospital, Chang Gung University College of Medicine, Kaohsiung 83301, Taiwan; 5Institute for Translational Research in Biomedicine, Kaohsiung Chang Gung Memorial Hospital, Kaohsiung 83301, Taiwan; 6School of Medicine, College of Medicine, Chang Gung University, Taoyuan 33302, Taiwan; 7Division of Urology, Department of Surgery, Kaohsiung Chang Gung Memorial Hospital, Chang Gung University College of Medicine, Kaohsiung 83301, Taiwan

**Keywords:** bladder cancer cells, cellular prion protein, cell signaling pathway, melatonin, cisplatin

## Abstract

This study investigated whether melatonin (Mel) would promote cisplatin to suppress the proliferation and growth of bladder cancer (BC) cells by inhibiting cellular prion protein (PrP^C^)-mediated cell stress and cell proliferation signaling. An immunohistochemical staining of tissue arrays from BC patients demonstrated that the PrP^C^ expression was significantly upregulated from stage I to III BC (*p* < 0.0001). The BC cellline of T24 was categorized into G1 (T24), G2 (T24 + Mel/100 μM), G3 (T24+cisplatin/6 μM), G4 (PrP^C^ overexpression in T24 (i.e., PrP^C-OE^-T24)), G5 (PrP^C-OE^-T24+Mel), and G6 (PrP^C-OE^-T24+cisplatin). When compared with a human uroepithelial cell line (SV-HUC-1), the cellular viability/wound healing ability/migration rate were significantly increased in T24 cells (G1) and further significantly increased in PrP^C-OE^-T24 cells (G4); and they were suppressed in Mel (G2/G5) or cisplatin (G3/G6) treatment (all *p* < 0.0001). Additionally, the protein expressions of cell proliferation (PI3K/p-Akt/p-m-TOR/MMP-9/PrP^C^), cell cycle/mitochondrial functional integrity (cyclin-D1/clyclin-E1/ckd2/ckd4/mitochondrial-cytochrome-C/PINK1), and cell stress (RAS/c-RAF/p-MEK1/2, p-ERK1/2) markers showed a similar pattern of cell viability among the groups (all *p* < 0.001). After the BC cell line of UMUC3 was implanted into nude mouse backs, by day 28 mthe BC weight/volume and the cellular levels of PrP^C^/MMP-2/MMP-9 were significantly, gradually reduced from groups one to four (all *p* < 0.0001). The protein expressions of cell proliferation (PI3K/p-Akt/p-m-TOR/MMP-9/PrP^C^), cell cycle/mitophagy (cyclin-D1/clyclin-E1/ckd2/ckd4/PINK1), and cell stress (RAS/c-RAF/p-MEK1,2/p-ERK1,2) signaling were significantly, progressively reduced from groups one to four, whereas the protein expressions of apoptotic (Mit-Bax/cleaved-caspase-3/cleaved-PARP) and oxidative stress/mitochondrial damaged (NOX-1/NOX-2/cytosolic-cytochrome-C/p-DRP1) markers expressed an opposite pattern of cell proliferation signaling among the groups (all *p* < 0.0001). Mel-cisplatin suppressed BC cell growth/proliferation via inhibiting the PrP^C^ in upregulating the cell proliferation/cell stress/cell cycle signaling.

## 1. Introduction

It is well known that urinary bladder cancer (UBC) is the second most common urologic malignancy [1]. Transitional cell carcinoma (TCC), which composes the greater proportion of UBC (estimated to be >95%), may develop into carcinoma in situ or invasive malignancy [2].

Currently, radical surgery with bilateral lymph node removal is recommended as the standardized treatment for the muscle invasion of urinary bladder cancer and accomplishes better regional control with very low rates of perioperative death (<1%) [3,4,5]. However, UBC is awfully invasive and often leads to mortality after distal metastasis [6]. In patients with distal metastasis, chemotherapy is the last-resort treatment. In fact, metastatic UBC is commonly a chemo-sensitive cancer, and cisplatin-based therapy is generally considered the standard of palliative treatment. However, despite the combined therapy (i.e., surgical intervention and chemotherapy), UBC remains a truculent malignancy with a high relapse rate [6], and the five-year survival rate of advanced UBC is approximately 15%. Thus, treatment of the advanced stage of UBC remains a challenge to physicians. Thus, an innovative treatment for improving the survival rate of UBC patients is of the utmost importance to both physicians and patients. However, prior to finding an effective therapeutic option for achieving our purposes, clarifying the underlying mechanism involved in the proliferation and progression of UBC should be the priority.

It is well-recognized that the cellular prior protein (PrP^C^) serves as a distinctive protein for neural protection and survival since this cellular protein impedes B-cell lymphoma 2 (Bcl-2)-associated, protein X (Bax)-regulated cell death [7]. In addition, fundamental investigations have further elucidated that PrP^C^-null cells are more vulnerable to serum denudation and oxidative stress than PrP^C^-express cells [8,9]. Interestingly, abundant data establish that PrP^C^ is involved in neoplasm-intrinsic behaviors, including proliferation, apoptosis, invasion, metastasis, and chemoresistance, as well as serving as a contributor for the self-renewal of cancer cells [10,11,12,13,14]. Additionally, PrP^C^ is also a contributive marker for the progression of colorectal carcinoma [15]. These reports raise our hypothesis: “could PrP^C^ be possible to play the key role of proliferation and progression of UBC?” Surprisingly, to the best of our knowledge, there are no surveyed reports that address the impact of PrP^C^ on these intrinsic behaviors of UBC.

Interestingly, one in vitro study previously demonstrated that PrP^C^ promoted the tumorigenesis and proliferation of gastric cancer cells via the upregulation of the PI3K/Akt pathway [16]. However, whether PrP^C^ also plays an essential role in the tumorigenesis and proliferation of human UBC by following the same signaling pathway has yet to be explored.

It is well known that the PI3K/Akt/m-TOR are two intracellular signaling pathways fundamental to numerous aspects of cell growth/the cell cycle and cell survival in physiological and pathological situations (e.g., cancer) [17,18]. Additionally, our previous study showed that a tissue plasminogen activator (tPA) enhanced the kinetics and circulating level of EPCs in the ischemic area by increasing the expression of matrix metalloproteinase (MMP)-9 [19]. Intriguingly, our previous investigation [20] identified that PI3K/Akt signaling regulated MMP-9 for enhancing the migration, invasion, and progression of UBC cells. 

Interestingly, melatonin (Mel), predominantly synthesized by the pineal gland and secreted into circulation in humankind, has been identified to act as a strong antioxidant [21,22] that suppresses generations of oxidative stress/free radicals and inflammatory reactions. Abundant data have proved that Mel protected against the development of cancer [23,24,25,26,27], principally through its distinctively oncostatic characteristics, including its ability to elicit apoptosis, vanquish oxidative stress, upregulate immunomodulation, and suppress angiogenesis and metastasis [28,29]. The distinctively important finding in our previous study was that Mel fundamentally downregulated the ZNF746-restrained bladder tumorigenesis by inhibiting the AKT-MMP-9 pathway in rodents [20]. However, when we examined the result of our study [20], we found that the tumor size was not totally diminished. Additionally, it is commonly identified that, whatever aggressive treatments have been taken, the majority of stage IV cancer could not be cured, suggesting that the malignancy could have more than one signaling pathway for its survival, growth, and distal metastasis. Based on the aforementioned issues [10,12,13,14,15,16,17,18,19,20], we conducted this study to elucidate the role of PrP^C^ in UBC cell survival/proliferation and invasion.

## 2. Results

### 2.1. Role of PrP^C^ on Regulating PI3K/Akt/m-TOR Pathway-Mediated Matrix Metalloproteinase 9 (Appendix A)

First, to assess the role of PrP^C^ in regulating the PI3K/Akt/m-TOR-MMP9 signaling, we categorized the cells into four groups, i.e., control (SV-HUC-1), S1 (T24 cells), S2 (PrP^C-OE^-T24 cells), S3 (siRNA-PrP^C^ in T24 cells, i.e., T24 cells with PrP^C^ silencing), and S4 (PrP^C-OE^-T24 cells + LY294002 (20 μM), i.e., a PI3K inhibitor) prior to cell culturing. The results demonstrated that, when compared with the control, the protein levels of PI3K, p-Akt, p-m-TOR, and MMP9 were remarkably increased in S1 and more remarkably increased in S2, whereas these parameters were substantially reduced in S3 and S4, suggesting that PrP^C^ played a crucial role in upregulating this cell proliferation signaling (Appendix A).

Second, to elucidate whether Mel and cisplatin therapies would inhibit the cell proliferation signaling in T24 cells, we categorized this BC cell line into T1 (i.e., T24 cells only), T2 (T24 cells + Mel (100 μM) for 24 h co-culture), and T3 (T24 cells + cisplatin (6 μM) for 24 h co-culture). The results demonstrated that the protein levels of PI3K, p-Akt, p-m-TOR, and MMP-9 were notably suppressed in T2 and more notably suppressed in T3, whereas the protein level of P-TEN, an index of anti-cancer proliferation, revealed an inversed manner of the PI3K/p-Akt/p-m-TOR biomarkers among the groups (Appendix A), suggesting the results supported our hypothesis.

### 2.2. The Cellular Expression of PrP^C^ in Different Stages of Human Bladder Carcinoma (Figure 1)

We hypothesized that there could be a strong correlation between an increased intensity of cellular expression of PrP^C^ and a progressive increase in BC stages in urinary bladder cancer patients. Therefore, we acquired a tissue array from US Biomax Inc., and performed an IHC staining. The analytical results illustrated that (1) the cellular expression of PrP^C^ in BC tissue was significantly increased when compared with the normal bladder specimen; (2) when we looked at the correlation between BC stage (i.e., TNM staging system) and the expression of PrP^C^, we found that the cellular expression of PrP^C^ was significantly upregulated in patients with stage II BC when compared to those with Stage I BC, suggesting that our hypothesis was reasonable. However, for the tissue array specimens, we could only obtain two patients with stage III BC and no stage IV patient who received an operation to provide a BC specimen (i.e., due to advanced BC that was not suitable for surgical intervention). Accordingly, we had no adequate sample sizes of stages III and IV of BC for statistical analysis. (3) When we examined the tumor grading and the *TNM staging* system, we also found that there was a strong correlation among these two BC classifications and the expression of PrP^C^; (4) two interesting findings were observed during analysis. First, we identified that the PrP^C^ expression in nuclei was remarkably increased compared to its expression in the cytoplasm of BC cells. Second, we also found that the expression of PrP^C^ intensity was significantly enhanced in the elder patients than in the younger patients. To the best of our knowledge, the significance of these two findings has yet to be explored. Our findings from the tissue array may suggest that the PrP^C^ might correlate with BC growth and proliferation (Figure 1).

**Figure 1 ijms-24-03353-f001:**
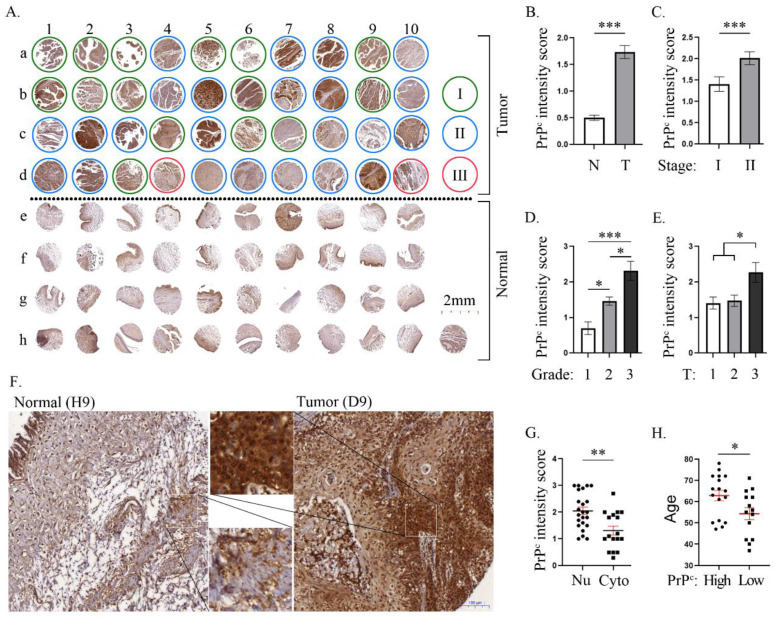
**The cellular expression of PrP^C^ in human bladder cancer patients.** (**A**) Immunohistochemical results of cellular prion protein (PrP^C^) on bladder cancer (BC) tissue arrays from 38 BC patients. Columns a to d represented harvested BC tissue; e to f represented normal specimens (i.e., normal part of BC), i.e., indicating that (a) represented a BC specimen and (d) corresponded to a normal specimen from the same BC patient. The green, blue, and red circles represent Stage I, II, and III classifications (i.e., TNM staging system), respectively. (**B**) The expression of PrP^C^ in the tumor part was remarkably increased when compared to the normal part from the same patient. *** represents the *p* value < 0.001 by paired *t*-tests. N = normal; T = tumor. (**C**) The expression of PrP^C^ in Stage II was significantly more enhanced than the Stage I in the tumor part. *** represents the *p* value < 0.001 by unpaired *t*-tests. (**D**) The expression of PrP^C^ was notably and gradually increased from grade 1 to grade 3. * and *** represent the *p* value < 0.05 and <0.001 by one-way ANOVA analysis. (**E**) The expression of PrP^C^ in Stage 3 (T stage in TNM classifications) was significantly stronger than in Stages I and II. * represents the *p* value < 0.05 by one-way ANOVA analysis. (**F**) Immunohistochemical result of PrP^C^ identified that the nuclear translocation in PrP^C^ is strongly expressed in the tumor part rather than in the normal part of the same patient. (**G**) The PrP^C^ expression was remarkably increased in nuclear (Nu) translocation rather than in cytoplasm (Cyto). ** represents the *p* value < 0.01 in unpaired *t*-tests. (**H**) The expression of PrP^C^ was significantly increased in older patients when compared to younger ones. The classifications of high vs. low PrP^C^ expression were separated by mean ± 1 S.E.M. * represents the *p* value < 0.05 by unpaired *t*-tests.

### 2.3. The Cell Viability and Wound Healing Rate (Figure 2)

To test the impact of PrP^C^ overexpression (i.e., *PRNP* gene overexpression in T24 cells, denoted as PrP^C-OE^-T24), Mel, and cisplatin on T24 cell viability, the MTT assay was adopted in the present study. The result showed that, when compared with the control (i.e., SV-HUC-1), the cell viability was notably increased in T24 (G1) cells at the time intervals of 24/48/72 h for the cell culture. However, this parameter was significantly inverted by Mel (G2) and more remarkably inverted by cisplatin (G3) at these time points. When looked at the situation of PrP^C^ overexpression in T24 cells (i.e., PrP^C-OE^-T24), we found that the cell viability was significantly increased in PrP^C-OE^-T24 cells (G4) when compared to SV-HUC-1 and G1 to G3 at all time intervals of 24, 48, or 72 h, indicating that the condition of PrP^C-OE^ would accelerate the speed of cell proliferation. The important finding was that this cell viability was significantly suppressed by Mel (G5) and more significantly suppressed by cisplatin (G6).

To test the wound-healing process, i.e., an indicator of cell proliferation, growth, and expansion rate in the above-mentioned conditions, the cells were incubated for 24 h. As we expected, the result of the wound-healing process exactly followed a similar pattern of cell viability among the SV-HUC-1 and G1 to G6 groups (Figure 2).

**Figure 2 ijms-24-03353-f002:**
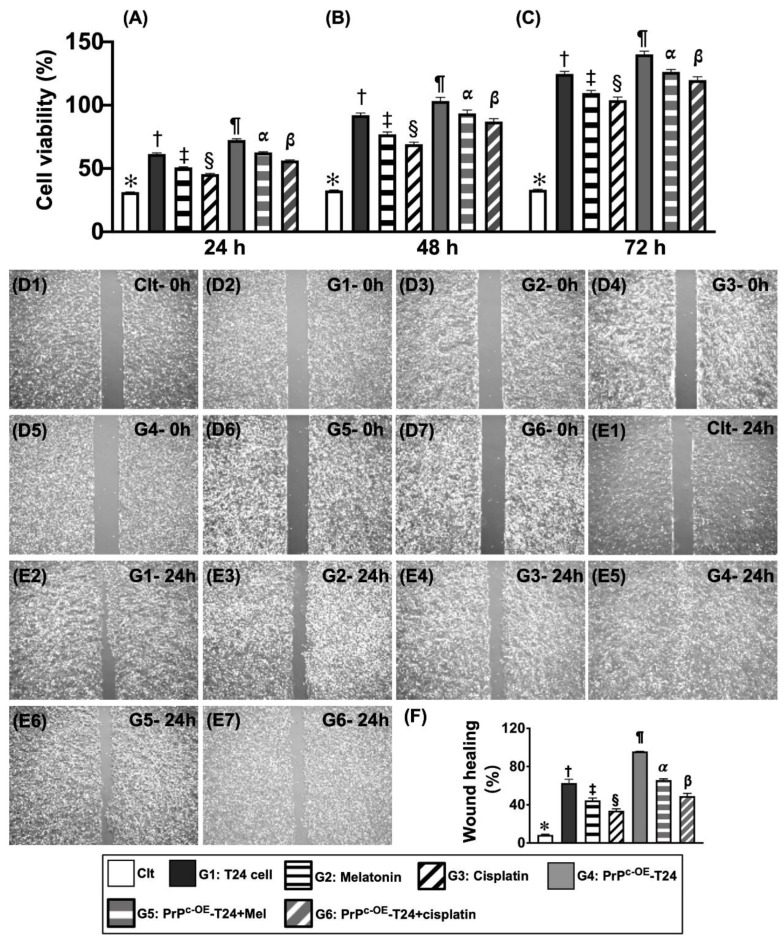
**The cell viability and wound healing rate in different situations.** (**A**) Analysis of the cell survival rate by 24 h (i.e., cell viability), * vs. different symbols (†, ‡, §, ¶, α, and β), *p* < 0.0001. (**B**) Analysis of the cell survival rate by 48 h (i.e., cell viability), * vs. different symbols (†, ‡, §, ¶, α, and β), *p* < 0.0001. (**C**) Analytical result of the cell survival rate by 72 h (i.e., cell viability), * vs. different symbols (†, ‡, §, ¶, α, and β), *p* < 0.0001. (**D1**–**D4**,**E1**–**E7**) The microscopic findings (40×) at 0 h (i.e., (**D1**–**D7**)) and 24 h (i.e., (**E1**–**E7**)) for illustrating the “wound healing” process (cell amount: 5.0 × 10^4^/well) between the seven groups. (**F**) Analysis of percentage of wound healing at 24 h, * vs. different symbols (†, ‡, §, ¶, α, and β), *p* < 0.0001. Symbols: (*, †, ‡, §, ¶, α, and β) or letters (**E1**–**E7**) indicate significance for each other (at 0.05 level). All statistical analyses were conducted by one-way ANOVA, followed by Bonferroni multiple comparison post hoc test (n = 6 for each group). The formula for the calculation of the wound healing process (%) = cell migrated area at 24 h/the original migrated area at 0 h. PrP^C^ = cellular prion protein. Control (Clt) = V-HUC-1; G1 = T24 only; G2 = melatonin (Mel); G3 = cisplatin; G4 = PrP^C^ overexpression in T24 cells (i.e., PrP^C-OE^-T24); G5 = PrP^C-OE^-T24 cells + Mel (100 μM treated for 24 h); G6 = PrP^C-OE^-T24 cells + cisplatin (6.0 μM treated for 24 h).

### 2.4. Migratory Capacity and Colony Formation Unit (CFU) of T24 BC Cell Line (Figure 3)

To elucidate whether Mel and cisplatin treatments could inhibit colony formation and the migration of BC cells, the BC cell line of T24 cells and conditions mentioned in Figure 2 were employed in our in vitro investigation. The result illustrated that, when compared with SV-HUC-1, the migratory ability and CFUs were significantly increased in G1 at the time point of the 24 h cell culture. On the other hand, these parameters (i.e., migratory ability and CFUs) were significantly reversed in G2 and more significantly reversed in G3 at the time interval of 24 h for the cell culture. Additionally, when we utilized the PrP^C-OE^-T24 cells for keen investigation, we further identified that these two parameters were significantly upregulated in PrP^C-OE^-T24 cells (G4) when compared to those in SV-HUC-1 and G1 to G3, implying that the condition of PrP^C-OE^ would accelerate the speed of migratory ability and CFUs. Of note is that the migratory ability and CFUs were substantially restrained by Mel (G5) and more substantially restrained by cisplatin (G6) (Figure 3).

**Figure 3 ijms-24-03353-f003:**
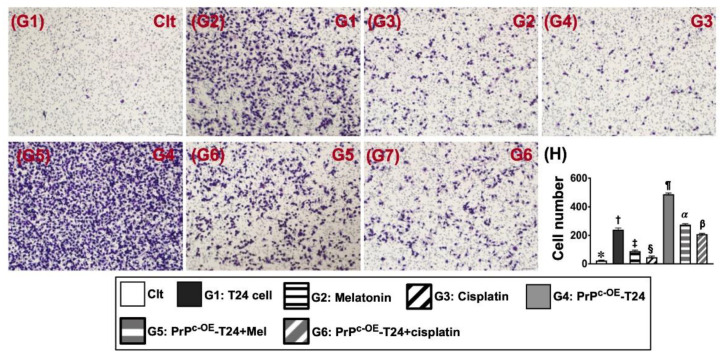
**Migratory ability of T24 cells in different conditions.** (**G1**–**G7**) Showing the microscopic findings (100×) for identifying the migratory T24 cells (gray color). (**H**) Analysis of the number of migratory cells, * vs. different symbols (†, ‡, §, ¶, α, and β), *p* < 0.0001. n = 5 for each group. Control (Clt) = V-HUC-1; G1 = T24 only; G2 = melatonin (Mel); G3 = cisplatin; G4 = PrP^C^ overexpression in T24 cells (i.e., PrP^C-OE^-T24); G5 = PrP^C-OE^-T24 cells + Mel (100 μM treated for 24 h); G6 = PrP^C-OE^-T24 cells + cisplatin (6.0 uM treated for 24 h).

### 2.5. Cell Proliferation, Cell Stress, and Cell Cycle Signaling, Oxidative Stress, and Cell Apoptosis (Figure 4, Figure 5, Figure 6 and Figure 7)

In addition to an assessment of the impacts of PrP^C^, Mel, and cisplatin on the T24 cellular-level expressions, we also investigated the impact of these three factors on the protein expressions of the signaling of T24 cells in the same situations, i.e., the cells were categorized from G1 to G6.

The result of the Western blot analysis revealed that the protein levels of PrP^C^, PI3K, p-Akt, and p-m-TOR, four cell-proliferation biomarkers, and MMP-9, an indicator of a cell migratory helper, were significantly increased in G1 when compared to the control (i.e., SV-HUC-1) (Figure 4). However, these biomarkers were remarkably inverted in G2 and more remarkably inverted in G3 (Figure 4). Additionally, we further recognized that these biomarkers were significantly upregulated in G4 (i.e., PrP^C-OE^-T24 cells) than in those of the control and G1 to G3, implying that the condition of PrP^C-OE^ would upregulate the cell-survival/proliferative signaling (Figure 4). Of notable significance was that these parameters were notably suppressed in G5 (i.e., by Mel treatment) and were more significantly suppressed in G6 (i.e., by cisplatin treatment) (Figure 4).

**Figure 4 ijms-24-03353-f004:**
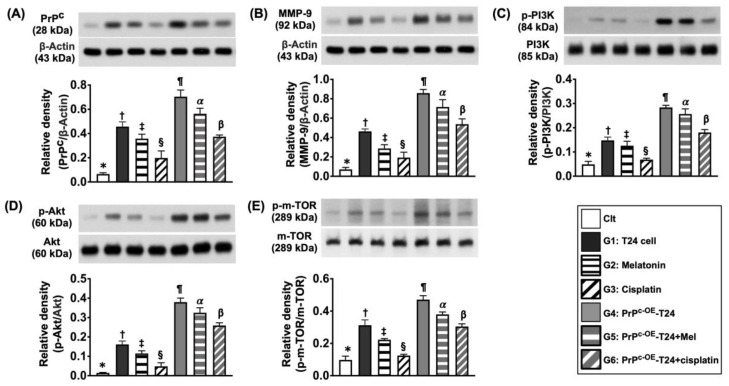
**The protein levels of cell proliferation markers in different situations.** (**A**) Protein level of PrP^C^: (1) * vs. different symbols (†, ‡, and §) and (2) ¶ vs. different symbols (α and β), all *p* values < 0.0001. (**B**) Protein level of phosphorylated (p)-PI3K: (1) * vs. different symbols (†, ‡, and §) and (2) ¶ vs. different symbols (α and β), all *p* values < 0.0001. (**C**) Protein level of p-Akt: (1) * vs. different symbols (†, ‡, and §) and (2) ¶ vs. different symbols (α and β), all *p* values < 0.0001. (**D**) Protein level of p-m-TOR: (1) * vs. different symbols (†, ‡, and §) and (2) ¶ vs. different symbols (α and β), all *p* values < 0.0001. (**E**) Protein level of MMP-9: (1) * vs. different symbols (†, ‡, and §) and (2) ¶ vs. different symbols (α and β), all *p* values < 0.0001. n = 6 for each group. Control (Clt) = V-HUC-1; G1 = T24 only; G2 = melatonin (Mel); G3 = cisplatin; G4 = PrP^C^ overexpression in T24 cells (i.e., PrP^C-OE^-T24); G5 = PrP^C-OE^-T24 cells + Mel (100 uM treated for 24 h); G6 = PrP^C-OE^-T24 cells + cisplatin (6.0 uM treated for 24 h).

Next, when we looked at the protein expressions of cyclin-D1, cyclin-E1, ckd2, ckd4, and the mitochondrial cytochrome, C, five biomarkers of cell cycle/mitochondrial functional integrity and the protein expression of PINK1, a mediator of mitophagy/fission (Figure 5), as well as the protein levels of RAS, c-RAF, p-MEK1/2, and p-ERK1/2 (four indices of cell stress biomarkers) (Figure 6), all exhibited a similar manner of cell proliferation signaling among the groups of SV-HUC-1 and G1 to G6.

**Figure 5 ijms-24-03353-f005:**
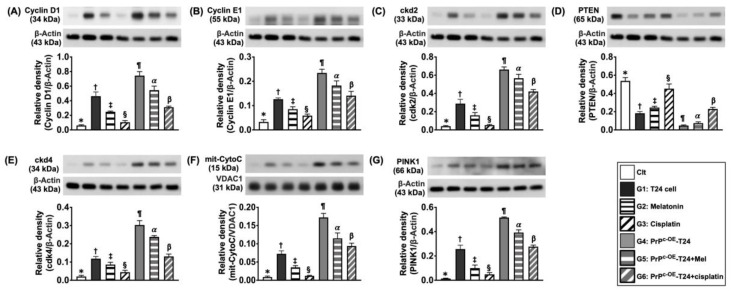
**The protein levels of cell cycle signaling in different situations.** (**A**) Protein level of cyclin D1: (1) * vs. different symbols (†, ‡, and §) and (2) ¶ vs. different symbols (α and β), all *p* values < 0.0001. (**B**) Protein expression of cyclin E1: (1) * vs. different symbols (†, ‡, and §) and (2) ¶ vs. different symbols (α and β), all *p* values <0.0001. (**C**) Protein level of ckd2: (1) * vs. different symbols (†, ‡, and §) and (2) ¶ vs. different symbols (α and β), all *p* values < 0.0001. (**D**) Protein level of phosphatase and tensin homolog (p-TEN): 1) * vs. different symbols (†, ‡, and §) and (2) ¶ vs. different symbols (α and β), all *p* values <0.0001. (**E**) Protein level of ckd4: (1) * vs. different symbols (†, ‡, and §) and (2) ¶ vs. different symbols (α and β), all *p* values <0.0001. (**F**) Protein level of mitochondrial cytochrome C (mit-CytoC): (1) * vs. different symbols (†, ‡, and §) and (2) ¶ vs. different symbols (α and β), all *p* values <0.0001. (**G**) Protein level of PTEN-induced kinase 1 (PINK1): (1) * vs. different symbols (†, ‡, and §) and (2) ¶ vs. different symbols (α and β), all *p* values <0.0001. n = 6 for each group. Control (Clt) = V-HUC-1; G1 = T24 only; G2 = melatonin (Mel); G3 = cisplatin; G4 = PrP^C^ overexpression in T24 cells (i.e., PrP^C-OE^-T24); G5 = PrP^C-OE^-T24 cells + Mel (100 uM treated for 24 h); G6 = PrP^C-OE^-T24 cells + cisplatin (6.0 uM treated for 24 h).

**Figure 6 ijms-24-03353-f006:**
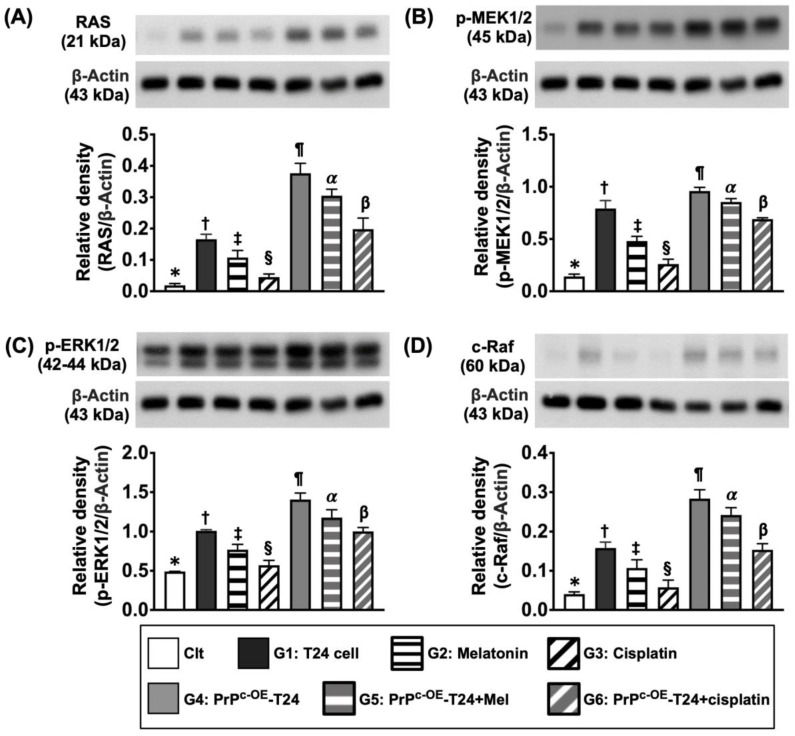
**Protein expressions of cell stress signaling in different situations.** (**A**) Protein level of RAS: (1) * vs. different symbols (†, ‡, and §) and (2) ¶ vs. different symbols (α and β), all *p* values < 0.0001. (**B**) Protein level of p-MEK1/2): (1) * vs. different symbols (†, ‡, and §) and (2) ¶ vs. different symbols (α and β), all *p* values < 0.0001. (**C**) Protein level of phosphorylated extracellular signal-regulated kinase 1 and 2 (p-ERK1/2): (1) * vs. different symbols (†, ‡, and §) and (2) ¶ vs. different symbols (α and β), all *p* values < 0.0001. (**D**) Protein level of proto-oncogene c-Raf (c-Raf): (1) * vs. different symbols (†, ‡, and §) and (2) ¶ vs. different symbols (α and β), all *p* values < 0.0001. n = 6 for each group. Control (Clt) = V-HUC-1; G1 = T24 only; G2 = melatonin (Mel); G3 = cisplatin; G4 = PrP^C^ overexpression in T24 cells (i.e., PrP^C-OE^-T24); G5 = PrP^C-OE^-T24 cells + Mel (100 μM treated for 24 h); G6 = PrP^C-OE^-T24 cells + cisplatin (6.0 uM treated for 24 h).

Further, when we inspected the protein expressions of NOX-1 and NOX-2 (Figure 7), two indices of oxidative stress, and the protein levels of mitochondrial Bax, cleaved (c)-PARP, and c-caspase3 (Figure 7), three indices of apoptosis, all revealed an identical manner of cell stress biomarkers among the groups.

**Figure 7 ijms-24-03353-f007:**
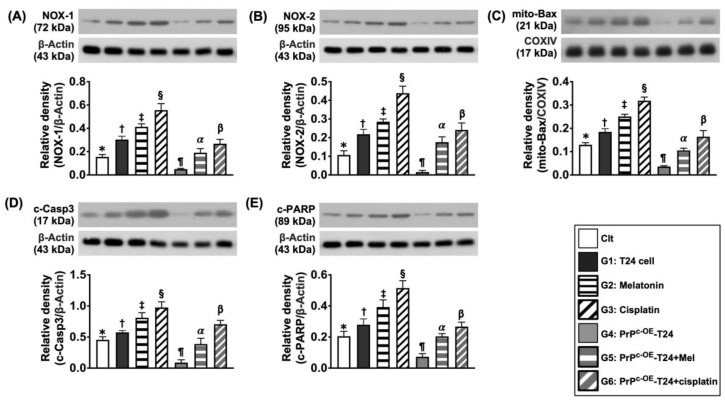
**Protein levels of oxidative stress and apoptosis in different situations.** (**A**) Protein level of NOX-1: (1) * vs. different symbols (†, ‡, and §) and (2) ¶ vs. different symbols (α and β), all *p* values < 0.0001. (**B**) Protein level of NOX-2: (1) * vs. different symbols (†, ‡, and §) and (2) ¶ vs. different symbols (α and β), all *p* values <0.0001. (**C**) Protein level of mitochondrial Bax (mit-Bax): (1) * vs. different symbols (†, ‡, and §) and (2) ¶ vs. different symbols (α and β), all *p* values <0.0001. (**D**) Protein level of cleaved (c) caspase 3 (c-Casp3): (1) * vs. different symbols (†, ‡, and §) and (2) ¶ vs. different symbols (α and β), all *p* values < 0.0001. (**E**) Protein expression of c-PARP: (1) * vs. different symbols (†, ‡, and §) and (2) ¶ vs. different symbols (α and β), all *p* values <0.0001. n = 6 for each group. Control (Clt) = V-HUC-1; G1 = T24 only; G2 = melatonin (Mel); G3 = cisplatin; G4 = PrP^C^ overexpression in T24 cells (i.e., PrP^C-OE^-T24); G5 = PrP^C-OE^-T24 cells + Mel (100 uM treated for 24 h); G6 = PrP^C-OE^-T24 cells + cisplatin (6.0 uM treated for 24 h).

### 2.6. Serial Changes of Tumor Volume and Tumor Weight at Day 28 after UMUC3 Bladder Cancer Cell Line Implantation into Nude Mouse Backs (Figure 8)

To evaluate the implanted tumor development in the living nude mice, the assessment of tumor dimensions occurred on days 7, 14, 21, and 28 after UMUC3 cancer cell implantation into the left and right backs of the nude mice. The results revealed that the total tumor volume (i.e., the summation of the left and right side) was notably gradually increased in group 1 (UMUC3 cells) from day 7 to day 28 after UMUC3 cell implantation. Additionally, by days 7 and 14 after UMUC3 cell implantation, the tumor volume showed no difference among the four groups. However, by days 21 and 28 after UMUC3 cell implantation, the tumor volume was remarkably diminished in group 2 (UMUC3 cell + Mel), further remarkably diminished in group 3 (UMUC3 cells + cisplatin) and further remarkably diminished in group 4 (UMUC3 cells + Mel + cisplatin) in these two time intervals when compared to group 1. Additionally, by day 28 after the tumor cell implantation, the harvested tumor weight displayed an identical pattern of day 28 tumor volume among the four groups. Our findings implied that the combined therapy of Mel and cisplatin offers the greatest benefit for suppressing the tumor growth (Figure 8).

**Figure 8 ijms-24-03353-f008:**
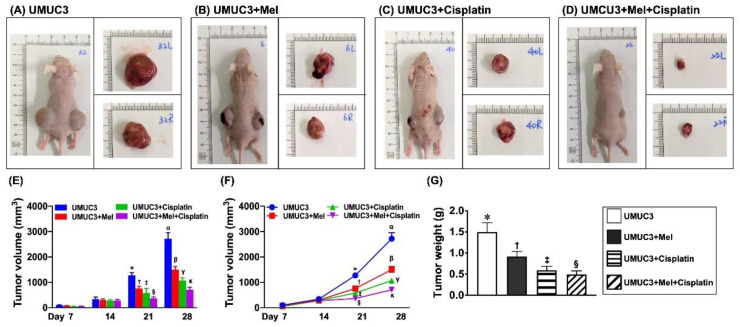
Serial changes of tumor volume and tumor weight on day 28 after UMUC3 bladder cancer cell engraftment into nude mouse backs. (**A**–**D**) Showing the grossly anatomical appearance of tumor masses in UMUC3 (**A**), UMUC3 + Mel (20 mg/kg/day) (**B**), UMUC3 + cisplatin (1 mg/kg/day) (**C**) UMUC3 + combined Mel-cisplatin (**D**) groups. The tumor size was remarkably diminished in UMUC3 +Mel, further remarkably diminished in UMUC3 + cisplatin and furthermore remarkably diminished in UMUC3 + combined Mel–cisplatin when compared to UMUC3 only by days 14, 21, and 28, respectively. (**E**) The analyses of the time points of tumor volume on days 7, 14, 18, 21, and 28. By days 7 and 14, the tumor volume did not differ among the four groups. On day 21, * vs. different symbols (†, ‡, and §), *p* < 0.0001. On day 28, α vs. different symbols (ß, γ, and κ), *p* < 0.0001. (**F**) Revealing the time courses of the growth of tumor volume size. The analytical results on the time intervals of days 7, 14, 18, 21, and 28 were exactly identical to (**E**). (**G**) On day 28, the tumor weight, * vs. different symbols (†, ‡, and §), *p* < 0.0001. n = 10 for each group. Mel = melatonin.

### 2.7. Cellular Expressions of MMPs and PrP^C^ in Harvested Tumors on Day 28 after UMUC3 Cell Engraftment into the Mouse Backs (Figure 9)

It is well-known that activation of proteolytic enzymes such as matrix metalloproteinases (MMPs) for cleavage of extra-cellular matrix (ECM) serves as the cardinal role for the invasion and migration of malignancy [30]. In the present study, we observed that as to compare with the group 1, the expressions of MMP-2 and MMP-9, two indices of MMPs for ECM proteolysis/degradation, were remarkably reduced in group 2, further remarkably reduced in group 3 and furthermore remarkably reduced in group 4. Additionally, the cellular expression of PrP^C^ illustrated a similar manner of MMPs between the groups (Figure 9).

**Figure 9 ijms-24-03353-f009:**
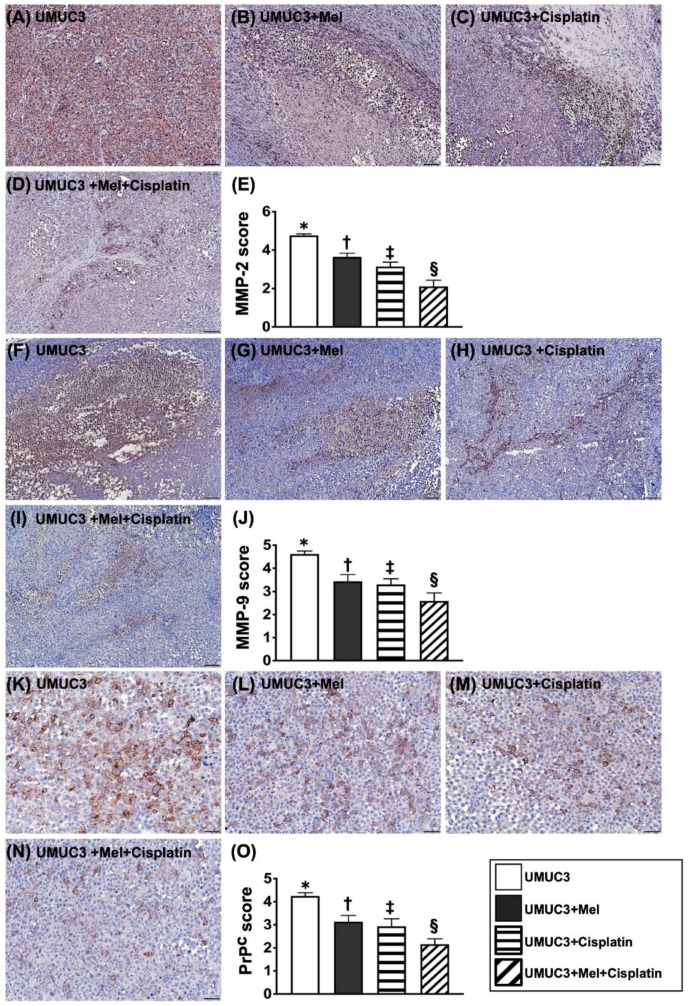
Cellular expressions of MMPs and PrP^C^ in harvested tumors on day 28 after UMUC3 cell engraftment into nude mouse backs. (**A**–**D**) Demonstrating the microscopic finding (400×) of IHC staining for identifying MMP-2 (gray color) expression. (**E**) Analysis of expression of intensity (i.e., scores) of positively stained MMP-2, * vs. different symbols (†, ‡, and §), *p* < 0.0001. (**F**–**I**) Showing the microscopic finding (400×) of IHC staining for identifying MMP-2 (gray color) expression. (**J**) Analysis of expression of intensity (i.e., scores) of positively stained MMP-9, * vs. different symbols (†, ‡, and §), *p* < 0.0001. (**K**–**N**) Demonstrating the microscopic finding (400×) for identifying cellular prion protein (PrP^C^) expression (gray color). (**O**) Analysis of expression of intensity (i.e., scores) of positively stained PrP^C^, * vs. different symbols (†, ‡, and §), *p* < 0.0001. Scale bars in right lower corner represent 20 µm. n = 8 for each group. Mel = melatonin.

### 2.8. The Protein Levels of Cell Survival/Cell Proliferation, Cell Stress and Oxidative-Stress/Mitochondrial Autophagic Biomarkers on Day 28 after UMUC3 Cell Engraftment into Nude Mouse Backs (Figure 10, Figure 11, Figure 12 and Figure 13)

After harvesting the tumor specimens, we employed a Western blot analysis to clarify the amounts of the three signaling biomarkers. The results of the Western blot analysis showed that the protein levels of p-PI3K, p-Akt, p-m-TOR, MMP-9, and PrP^C^, five indicators of cell proliferation, were remarkably and gradually reduced from groups 1 to 4 (Figure 10).

**Figure 10 ijms-24-03353-f010:**
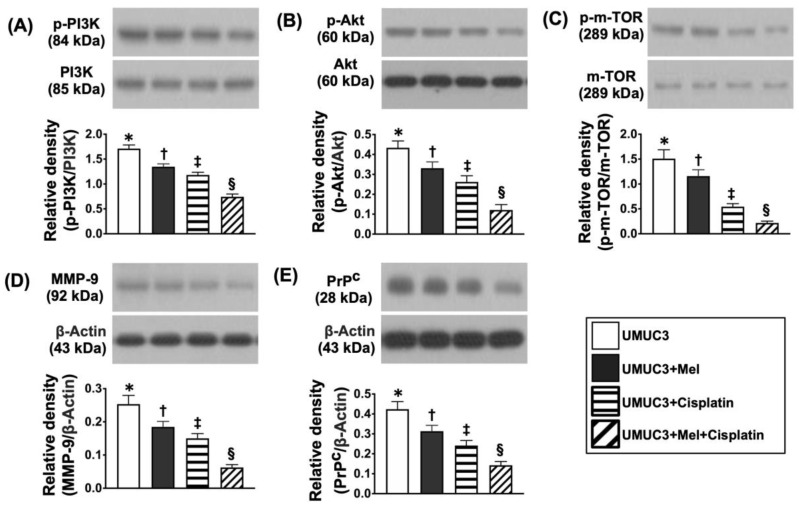
Protein expressions of cell survival/cell proliferation signaling on day 28 after UMUC3 engraftment into nude mouse backs. (**A**) Protein level of phosphorylated (p)-PI3K, * vs. different symbols (†, ‡, and §), *p* < 0.0001. (**B**) Protein level of p-Akt, * vs. different symbols (†, ‡, and §), *p* < 0.0001. (**C**) Protein level of p-m-TOR, * vs. different symbols (†, ‡, and §), *p* < 0.0001. (**D**) Protein level of matrix metalloproteinase (MMP)-9, * vs. different symbols (†, ‡, and §), *p* < 0.0001. (**E**) Protein level of PrP^C^, * vs. different symbols (†, ‡, and §), *p* < 0.0001. n = 6 for each group. Mel = melatonin.

Additionally, the Western blot analysis demonstrated that the protein levels of cyclin-D1, clyclin-E1, ckd2, and ckd4, four indices of cell cycle, and PINK1, an indicator of mitophagy, were also remarkably and gradually reduced from groups 1 to 4 (Figure 11). On the other hand, the protein level of P-TEN, an index of tumor suppression, revealed an inverse manner of cell cycling between the groups (Figure 11).

**Figure 11 ijms-24-03353-f011:**
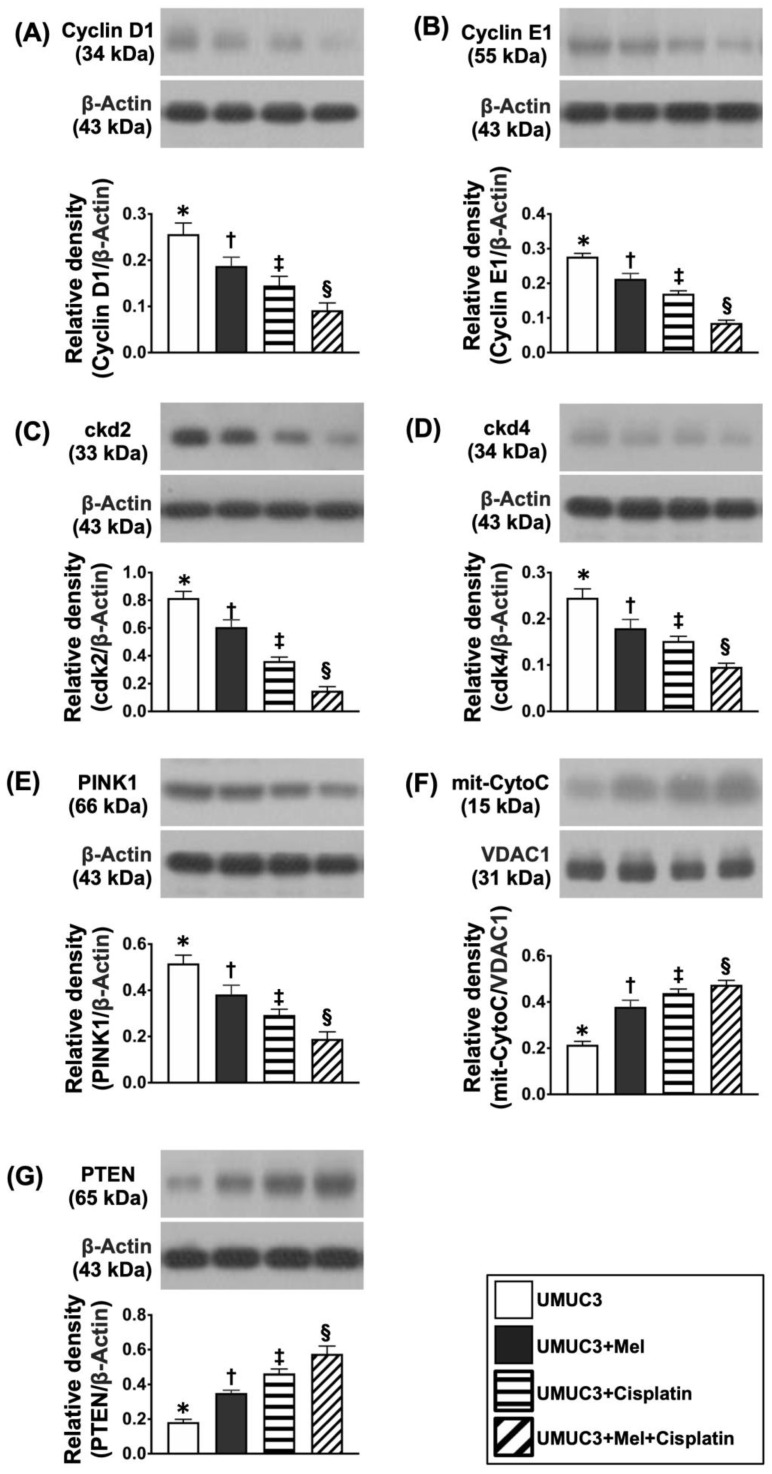
**Protein expressions of cell cycle biomarkers on day 28 after UMUC3 engraftment into nude mouse backs.** (**A**) Protein level of cyclin-D1, * vs. different symbols (†, ‡, and §), *p* < 0.0001. (**B**) Protein level of clyclin-E1, * vs. different symbols (†, ‡, and §), *p* < 0.0001. (**C**) Protein level of ckd2, * vs. different symbols (†, ‡, and §), *p* < 0.0001. (**D**) Protein level of ckd4, * vs. different symbols (†, ‡, and §), *p* < 0.0001. (**E**) Protein level of PINK1, * vs. different symbols (†, ‡, and §), *p* < 0.0001. (**F**) Protein level of mitochondrial cytochrome C (mit-CytoC), * vs. different symbols (†, ‡, and §), *p* < 0.0001. (**G**) Protein level for P-TEN, * vs. different symbols (†, ‡, §), *p* < 0.0001. n = 6 for each group. Mel = melatonin.

Furthermore, the Western blot analysis showed that that protein expressions of RAS, c-RAF, p-MEK1/2 and p-ERK1/2, four biomarkers of cell-stress signaling, displayed a similar pattern of cell cycle among the groups (Figure 12).

**Figure 12 ijms-24-03353-f012:**
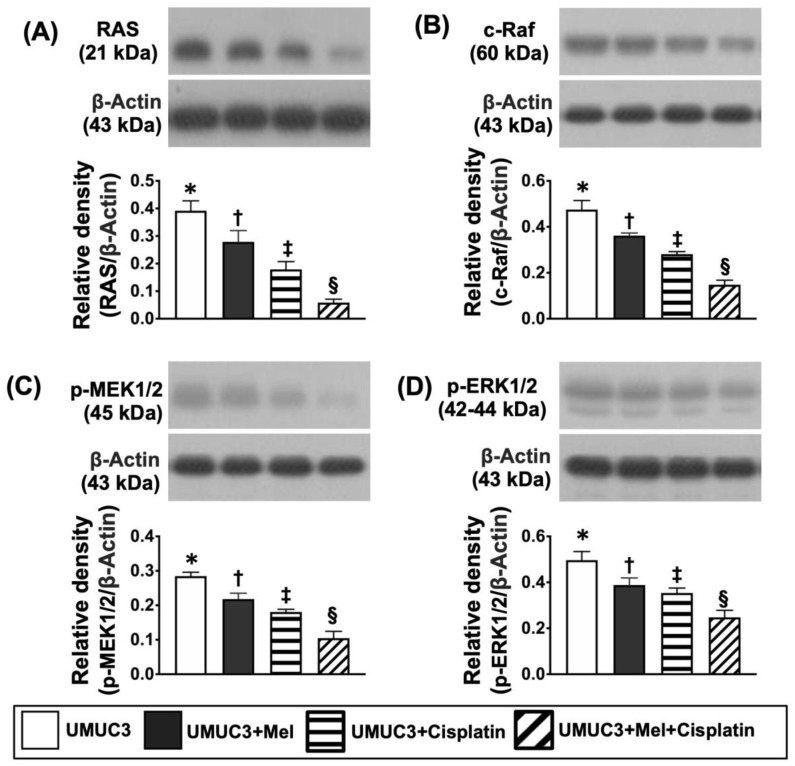
**Protein levels of cell stress signaling on day 28 after UMUC3 embedded into nude mouse back.** (**A**) Protein level of RAS, * vs. different symbols (†, ‡, and §), *p* < 0.0001. (**B**) Protein level of c-RAF, * vs. different symbols (†, ‡, and §), *p* < 0.0001. (**C**) Protein level of p-MEK1/2, * vs. different symbols (†, ‡, and §), *p* < 0.0001. (**D**) Protein level of p-ERK1/2, * vs. different symbols (†, ‡, and §), *p* < 0.0001. n = 6 for each group. Mel = melatonin.

On the other hand, the protein levels of mitochondrial Bax, c-caspase 3, and c-ARP, three indices of apoptosis, the protein levels of NOX-1 and NOX-2, two indicators of oxidative stress, and the protein level of p-DRP1, an index of fission (i.e., implicated mitochondrial damage biomarker) expressed an inverse manner of cell stress signaling (Figure 13).

**Figure 13 ijms-24-03353-f013:**
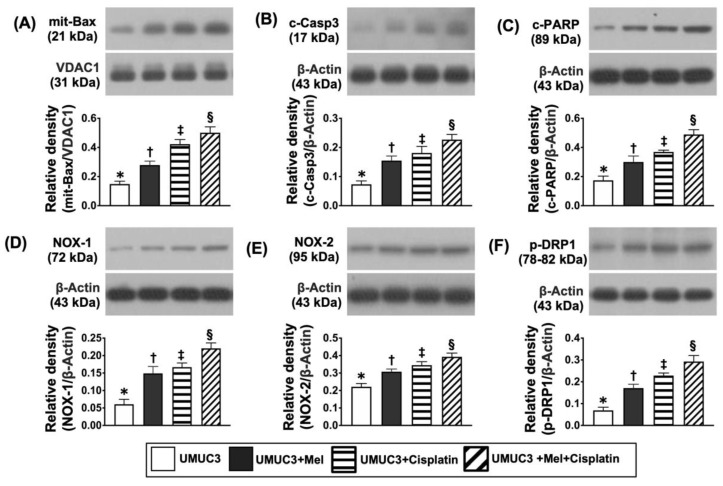
Protein levels of oxidative stress/mitochondrial damage apoptotic markers on day 28 after UMUC3 embedded into nude mouse back. (**A**) Protein level of mit-Bax, * vs. different symbols (†, ‡, and §), *p* < 0.0001. (**B**) Protein level of c-caspase 3 (c-Casp3), * vs. different symbols (†, ‡, and §), *p* < 0.0001. (**C**) Protein level of c-PARP, * vs. different symbols (†, ‡, and §), *p* < 0.0001. (**D**) Protein level of NOX-1, * vs. different symbols (†, ‡, and §), *p* < 0.0001. (**E**) Protein level of NOX-2, * vs. different symbols (†, ‡, and §), *p* < 0.0001. (**F**) Protein level of p-DRP1, * vs. different symbols (†, ‡, and §), *p* < 0.0001. n = 6 for each group. Mel = melatonin.

## 3. Discussion

This study, which investigated the role of PrP^C^ and the impacts of the pharmacomodulation of Mel and cisplatin on tumorigenesis and the growth of urinary BC cell lines, delivered several striking implications. First, the in vitro and in vivo studies. as well as the tissue array (i.e., derived from patients’ bladder cancer tissues), identified that PrP^C^ played a principal role in the proliferation, growth, and invasion of BC, mainly through upregulating the PI3K/Akt/m-TOR/MMPs signaling pathway. Second, Mel–cisplatin therapy remarkably suppressed the growth, invasion, and survival of the BC cells, predominantly through downregulating the cell survival/cell proliferation, oxidative cell stress, and cell cycle signaling pathways.

After realizing that PrP^C^ serves as a crucial neuroprotective protein [7], a cardinal role of cell proliferation/tissue regeneration [31], and an important biomarker for progression from colorectal adenoma to carcinoma [15], we innovatively proposed that the degree of the expression of PrP^C^ could be strongly correlated with the stage of UBC patients. Accordingly, we procured a tissue array (i.e., specimen collection solely from operable UBC patients, i.e., patients from stages I to III) for detailed analysis. As we expected, the ICH staining demonstrated that the cellular expression of PrP^C^ was strongly correlated with the cancer grade and tumor stage, implying that the results from the tissue array analysis favorably supported our hypothesis.

It is widely recognized that PI3K/Akt/m-TOR is the fundamental signaling pathway for cell survival/proliferation [17,18]. Additionally, our previous study showed that the PI3K-Akt-mTOR-MMPs signaling pathway acted as a primary role for promoting the migration, invasion, and progression of BC cells [20]. Furthermore, our recent study [32] revealed that PrP^C^ participated in the preservation of the residual renal function in rodents with CKD, mainly through the activation of PI3K-Akt-mTOR-MMPs signaling. These reports [17,18,20,32] encouraged and revitalized us to more thoroughly investigate the correlation between PrP^C^ and BC cell proliferation signaling. As expected, our in vitro study demonstrated that, when compared to the control (i.e., SV-HUC-1 cell line), the protein levels of PI3K-Akt-mTOR-MMP9 signaling were markedly upregulated in BC cells (i.e., T24 cell line), more upregulated in PrP^C-OE^-T24 cells, significantly suppressed in PrP^C^-silencing T24 cells, and also markedly downregulated in PrP^C-OE^-T24 cells treated with a PI3K inhibitor. As a result, our findings, in addition to being consistent with the findings of previous studies [17,18,20,32], proved that PrP^C^ was indispensable for promoting cell proliferation. Interestingly, the result of our in vitro study further revealed that, when compared with SV-HUC-1 cells, the cell viability, wound-healing rate, migratory capacity, and cell growth rate (i.e., colony formation unit) were substantially augmented in T24 cells and more significantly augmented in PrP^C-OE^-T24 cells. These findings once again proved that PrP^C^ serves as a fundamental factor for promoting the cell proliferation. It is of significance was that these upregulated parameters were substantially inhibited by Mel and more substantially inhibited by cisplatin treatment.

A previous study showed that cell stress signaling acted in a primary role on the proliferation, growth, and invasion of tongue squamous carcinoma cells [33]. Intriguingly, in our in vitro study, we further found that, when compared with the control, not only the protein level of cell proliferation signaling but also the protein expression of cell stress signaling (i.e., Figure 6) and cell cycle biomarkers (Figure 5) were markedly promoted in BC cells (i.e., the T24 cell line) and more markedly promoted in PrP^C-OE^-T24 cells. Accordingly, in addition to strengthening the finding from the previous study [33], our finding highlighted that PrP^C^ is also involved in regulating cell stress/cell cycle signaling (Figure 14 and Figure 15). However, the expression of these two signaling pathways weas remarkably suppressed in Mel and more remarkably suppressed in cisplatin treatment.

After obtaining the supporting evidence from the in vitro study and the clinical data, we performed the animal model study. The most important finding in the present in vivo study was that, when compared with the control (i.e., UMUC3 cell engraftment into the nude mouse backs), the tumor volume measured in living animals and the tumor weight measured after euthanizing the animals were significantly reduced following Mel therapy, more significantly reduced following cisplatin therapy, and further significantly reduced following the combined Mel–cisplatin treatment in nude mouse body, suggesting that when combined, Mel and cisplatin offered the greatest benefit in suppressing the tumor volume and tumor weight.

Readers would be highly interested in understanding the underlying mechanisms involved in the UBC proliferation and growth and the mechanistic basis of the Mel–cisplatin effective treatment. The principal finding from both the in vitro and in vivo studies was that at least three types of signaling, i.e., including cell proliferation, cell stress and cycle, and the mitochondrial integrity, were identified (refer to Figure 13 and Figure 14) to participate in the tumor cell proliferation and tumor growth. Of distinctive importance was that PrP^C^ was, once again, identified as playing a central role in the tumor proliferation and growth in the in vivo study. However, these molecular, cellular, and pathophysiological perturbations were significantly revised by the Mel–cisplatin treatment. In this way, we might provide a more complete, reliable, and satisfactory piece of information for our readers.

The enhancement of oxidative stress, apoptosis, and mitochondrial damage in the malignant neoplasm has been widely recognized in the phenomenon of any effective anti-cancer treatment [20,31,32]. In the present in vitro and in vivo studies, we found that the protein levels of oxidative stress, apoptosis, and mitochondrial damage in the tumor cell line and from the harvested tumors were markedly increased after Mel treatment, more markedly increased after cisplatin treatment and further markedly increased following combined Mel–cisplatin treatment. Our findings, in addition to corroborating with the findings of previous studies [20,31,32], could explain why the tumor volume and tumor mass were notably gradually reduced from Mel to combined Mel–cisplatin treatment.

### Study Limitations

This study has limitations. First, we did not test the dose-dependent effects of Mel and cisplatin. Thus, it might be inappropriate to conclude that the cisplatin was more effective than Mel or vice versa in suppressing the tumor growth and proliferation. Second, due to the fact that the study period was relatively short, this study did not investigate the possibility of distal metastasis in the UMUC3 cancer cells. Third, in the present study, we had only utilized two bladder cancer cell lines for the investigations’ serial molecular levels. However, to study the expression of PrP^C^ in human bladder cancer cell lines, a panel of cell lines should be examined to better convince the readers. In conclusion, the results of the present study demonstrated that PrP^C^ plays a crucial role in tumor growth and proliferation, mostly through upregulating the cell proliferation, cell stress, and cell cycle signaling, and that Mel–cisplatin therapy downregulated the PrP^C^-promoted expressions of these three signaling pathways.

## 4. Materials and Methods

### 4.1. Ethics Statement

All animal procedures were approved by the Institute of Animal Care and Use Committee at Kaohsiung Chang Gung Memorial Hospital (Affidavit of Approval of Animal Use Protocol No. 2018121104) and were performed in accordance with the Guide for the Care and Use of Laboratory Animals. 

### 4.2. Cell Culture

The bladder cancer (BC) cell lines T24 (Bioresource Collection and Research Center, Taiwan) and UM-UC-3 (i.e., UMUC3) (ATCC^®^ CRL-1749™ (American Type Culture Collection, Manassas, VA, USA)) and the normal bladder cell line of SV-HUC-1 (Bioresource Collection and Research Center, Hsinchu, Taiwan) were utilized in the present study.

### 4.3. Transfection of T24 BC Cell Line with Plasmids for Cellular Prion Protein (PrPC) Overexpression

The procedures and protocol for gene overexpression were reported in our recent study [31]. In detail, the prion protein expression vectors pCS6-*PRNP* and pEZ-M68 *PRNP* were purchased from Transomic Technologies (Huntsville, AL, USA.) and GeneCopoeia, Inc. (Rockville, MD, USA), respectively. These expression vectors were amplified by *E. coli* (DH5a strain).

The transient transfection of cells with plasmids was performed using Lipofectamine 3000 (Invitrogen, Life technologies, Carlsbad, CA, USA) according to the manufacturer’s instructions but with slight modifications. Cells were replated 24 h before transfection at a density of 1 × 10^6^ cells in 10 mL of fresh culture medium in a 10 cm plastic dish. For use in transfection, 30 μL of Lipofectamine 3000 was incubated with 15 μg of indicated expression vector at room temperature. The complex was incubated with cells at 37 °C in a humidified atmosphere of 5% CO_2_ before being harvested. To assess the drug resistance of PrP^C^ overexpressed by T24 cells (PRNP gene overexpression in T24 cells, denoted as PrP^C-OE^-T24), 2 × 10^4^ PrP^C-OE^-T24 cells were cultured in 96-well plates with and without Mel or cisplatin co-cultures. Following 24–96 h of incubation, the growth rate of the transfected T24 cells was detected using an MTT cell viability assay.

### 4.4. MTT Cell Viability Assay, Estimation of the Colony Formation and Wound-Healing Assay

The procedures and protocol were described in our previous report [34]. For the MTT assay, after 7 days of seeding, the cells were fixed and permeabilized with methanol. After washing the cells two times with PBS, the colony was observed using a microscope after a 10% Giemsa solution. 

### 4.5. Wound Healing Assessment

The procedures and protocol were based on our previous study [34]. Briefly, following overnight cell culturing, the migration was captured with microphotography. The total migratory distance was determined using Image J software.

### 4.6. Transwell Migration Assessment of the BC Cell Invasion Capacity

After 24 h of incubation, the nonmotile T24 cells at the top of the filter were removed and the motile cells at the bottom of the filter were fixed with methanol and stained with a one-tenth dilution of Giemsa. The number of migrated cells in each chamber was carefully counted in five randomly chosen fields.

### 4.7. Western Blot Analysis

The procedure and protocol were described in our recent reports [30,35]. In detail, equal amounts (30 µg) of protein extracts were separated by 8–12% SDS-PAGE. After electrophoresis, the separated proteins were transferred onto a polyvinylidene difluoride (PVDF) membrane (Amersham Biosciences, Amersham, UK). The immunoreactive membranes were visualized using enhanced chemiluminescence (ECL; Amersham Biosciences, Amersham, UK) and exposed to a medical X-ray film (FUJI).

### 4.8. Immunohistochemical (IHC) and Immunofluorescent (IF) Staining

The methodology was based on our recent reports [30,35]. Briefly, for IHC and IF staining, rehydrated paraffin sections were treated with 3% H_2_O_2_ for 30 min and incubated with Immuno-Block reagent for 30 min at room temperature. Sections were then incubated with primary antibodies, and sections incubated with the use of irrelevant antibodies served as controls.

### 4.9. Cell Grouping

In the in vitro study, the cell lines were divided into the control (SV-HUC-1 (healthy human uroepithelial cell line)), G1 (T24 cells only), G2 (T24 cells + Mel (100 μM) co-cultured for 24 h), G3 (T24 cells + cisplatin (6.0 μM) co-cultured for 24 h)), G4 (PrP^C-OE^-T24 cells only), G5 (PrP^C-OE^-T24 cells + Mel (100u μM) co-cultured for 24 h), and G6 (PrP^C-OE^-T24 cells + cisplatin (6.0 μM) co-cultured for 24 h)).

### 4.10. Animal Grouping, Treatment Strategy, and Assessment of Tumor Volume and Weight

Male BALB/c nude mice (n = 40) with a body weight of 20–22 g were divided into group 1 (UMUC3 cells (2.0 × 10^5^/100 μL), mixed with 100 μL of Matrigel, were embedded into nude mouse backs), group 2 (UMUC3 cells (2.0 × 10^5^/100 μL), mixed with 100μL of Matrigel, were embedded into nude mouse backs + Mel (20 mg/kg/day) by intraperitoneal administration from days 14 to 28 after UMUC3 cell implantation), group 3 (UMUC3 cells (2.0 × 10^5^/100 μL) + cisplatin (1 mg/kg/day) by intraperitoneal administration from days 14 to 24 after UMUC3 cell line implantation) and group 4 (UMUC3 cells (2.0 × 10^5^/100 μL) administration into nude mouse back + Mel + cisplatin).

The dosages of UMUC3 cells to be implanted [34] and the amounts of Mel [20,34], and cisplatin [36] to be used in this study were referred to our previous [20,34] and other previous [36] reports with minimal modification.

The UMUC3 cell growth in the living animals was quantitatively analyzed (neoplasm volume calculated as: width^2^ × length) at the time intervals of days 7, 14, 21, and 28 after engrafting the UMUC3 cells into the left and right backs of the nude mice. Additionally, the neoplasms were harvested quickly after the animals were euthanized on day 28, following the engraftment of UMUC3 cells for individual analysis.

### 4.11. Statistical Analysis

Variables were presented as the mean ± standard deviation. Statistics were conducted using ANOVA, followed by a Bonferroni multiple comparison post hoc test. SAS statistical software for Windows, version 8.2, was used for the analysis. *p* < 0.05 was viewed as statistically significant.

## Figures and Tables

**Figure 14 ijms-24-03353-f014:**
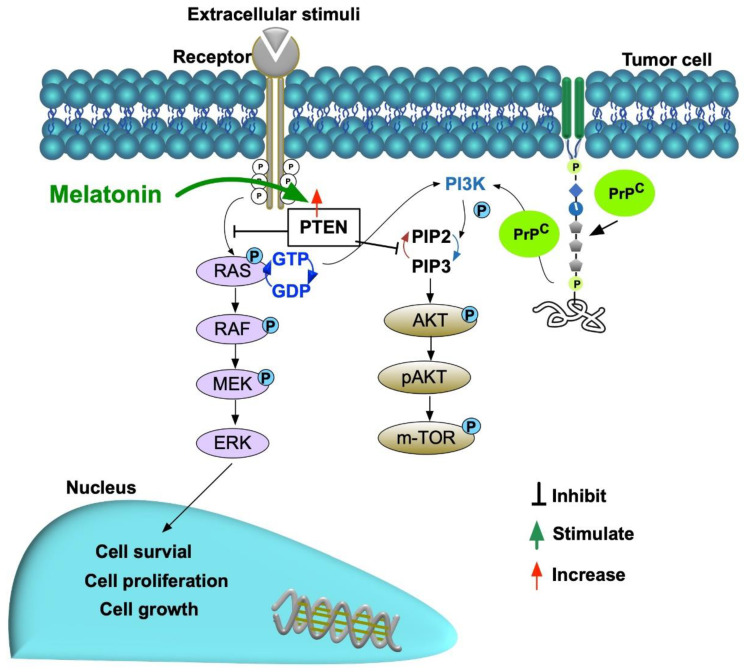
Schematically proposed mechanism of cell stress/cell proliferation signaling pathways (i.e., RAS and PI3K) involving in regulating the bladder cancer cell growth, differentiation, proliferation, and invasion and Melatonin upregulated the PTEN for suppressing RAS and PI3K signaling pathways. PrP^C^ = cellular prion protein.

**Figure 15 ijms-24-03353-f015:**
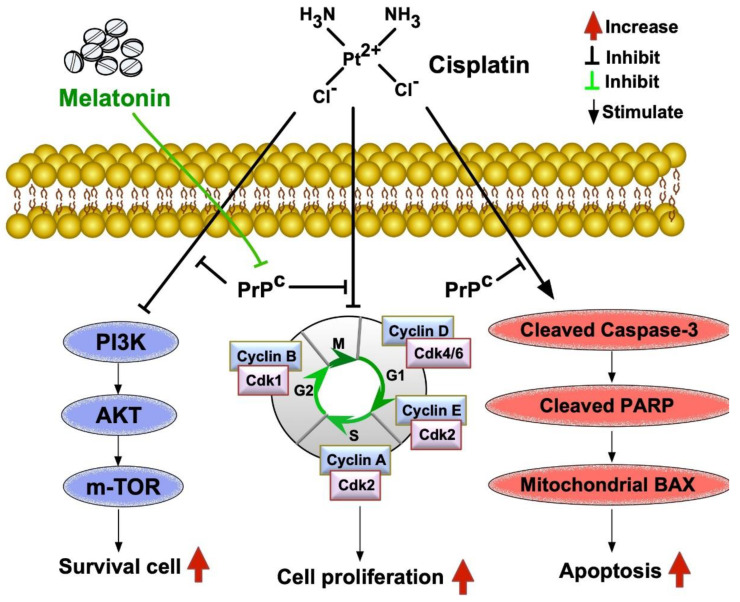
**Schematically proposed mechanism suggests Mel-supported cisplatin in the treatment of bladder cancer cells.** In this proposed mechanism, we suggest cisplatin effect would be attenuated by upregulation of PrP^C^ expression in bladder tumors that could be suppressed by melatonin (Me), implying the Mel-facilitated cisplatin effect.

## Data Availability

The dataset of the present study is available from the corresponding author upon request.

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
