# Peer review of "Melatonin-Assisted Cisplatin Suppresses Urinary Bladder Cancer Cell Proliferation and Growth through Inhibiting PrPC-Regulated Cell Stress and Cell Proliferation Signaling"

_ijms, 2023, doi:10.3390/ijms24043353_

Round 1
Reviewer 1 Report
In this work, the authors investigated the role of Melatonin (Mel) and cisplatin to suppress bladder-cancer (BC) cell proliferation and growth through inhibiting cellular prion protein (PrPC)-mediated cell-stress and cell-proliferation signaling. I have gone through the manuscript, and I found the topic and the work done of great interest, and suitable for publication in “International Journal of Molecular Sciences”. The work presented is diversified and includes many important results. I recommended the manuscript for publication in “International Journal of Molecular Sciences”
Author Response
We would like to thank you very much for your appreciation of our investigation!
Reviewer 2 Report
The authors examined the effects of melatonin on bladder cancer cell proliferation as a single agent or in combination with cisplatin.
They hypothesize that melatonin exerts its effects via inhibition of the cellular prion protein (PrPC) signaling. The authors performed a big amount of experiments to prove their hypothesis.
The main drawback of the manuscript is that the authors do not present a rationale for choosing melatonin as an inhibitor of PrPC signaling. It has been suggested that melatonin regulates cellular processes through PrPC-dependent pathways. On the other hand, melatonin has been reported to regulate a number of molecular pathways, including apoptosis, autophagy, etc. The authors do not even discuss in the introduction why they used melatonin in the present study.
If their objective was to investigate the combinatory treatment of cisplatin with melatonin, the effect (additive or synergistic) should be examined using isobolographic analysis.
Minor points.
1. Normal urothelial cell line SV-HUC-1 (established by transformation of normal ureter tissue) and urothelial bladder carcinoma cell line T24 were used for in vitro experiments and a single cell line UM-UC-3 was used for the xenograft tumor model. To study the expression of PrPC in human bladder cancer cell lines a panel of cell lines should be examined.
2. The authors should describe the transfection of T24 cells with PrPC in detail. Was the transfection transient or stable, what expression vector was used in the experiments? Did they use a viral vector?
3. Line 208. As far as I know, Duke's classification is not used in bladder cancer. It is used in colorectal cancer.
4. Line 222. “PrPC could play a cardinal role on BC growth and proliferation.” This conclusion cannot be drawn based on immunohistochemical staining done in this study.
5. It seems that it is incorrect to compare the intensity score of cytoplasmic staining with that of nuclear staining (Figure 1 G). Instead, the authors should compare the number of tumors with cytoplasmic staining and with nuclear staining.
6. The authors conclude that normal tissues had lower PrPC expression levels than bladder cancer tissues. However, in figure 1 F, normal urothelium is highly positive for PrPC. Nevertheless, the authors chose interstitial tissue to demonstrate low PrPC expression in this sample. Expression levels and staining patterns (nuclear or cytoplasmic) should be compared between bladder cancer tissues and normal urothelium and not other bladder wall layers.
7. Figure 2 A, B, and C could be arranged in a single line graph with time on the X-axis and cell viability on the Y-axis.
8. To further clarify PrPC function in bladder cancer cells, genetic depletion of PrPC using RNA interference should be used.
Author Response
Response to Reviewer 2# Comments
Dear Reviewer:
Your constructive criticism is greatly appreciated. We have made the following responses to comply with your honorable suggestions (Note: The revised parts of the manuscript in response to Reviewer’s comments have been marked in red color):
Response to Comments and Suggestions for Authors
Comment 1: The main drawback of the manuscript is that the authors do not present a rationale for choosing melatonin as an inhibitor of PrPC signaling. It has been suggested that melatonin regulates cellular processes through PrPC-dependent pathways. On the other hand, melatonin has been reported to regulate a number of molecular pathways, including apoptosis, autophagy, etc. The authors do not even discuss in the introduction why they used melatonin in the present study.
Response 1: Yes, according to your recommendation, we have discussed the rationale for why we utilized the melatonin in the present study in the Introduction paragraph of our revised manuscript.
Comment 2: If their objective was to investigate the combinatory treatment of cisplatin with melatonin, the effect (additive or synergistic) should be examined using isobolographic analysis
Response 2: Dear reviewer, the purpose of our study is to evaluate if combination strategy with cisplatin and melatonin is better than either one alone for suppressing the growth and proliferation of bladder cancer. In this study, we also intended to explore the possible mechanism of action of combination strategy in the cancer therapy. We used full-dose drugs for our test rather than effective doses of them, and therefore the detailed information of ED50 for both melatonin and cisplatin was unknown. Additionally, we did not test the (upper) 95% confidence limit of both drug dosages for the final outcome, so isobologram analysis cannot be carried out without the abovementioned parameters. Given your recommendation, we will test the additive and synergistic effects from these two drugs to provide more information about appropriate drug dose for this effective combination cancer treatment in our next study.
Response to minor points:
Point 1: Normal urothelial cell line SV-HUC-1 (established by transformation of normal ureter tissue) and urothelial bladder carcinoma cell line T24 were used for in vitro experiments and a single cell line UM-UC-3 was used for the xenograft tumor model. To study the expression of PrPC in human bladder cancer cell lines a panel of cell lines should be examined.
Response 1: Thank you for your professional comment. In fact, in the present study, we had analyzed two cancer cell lines, that is already fitted the minimal requirement of scientific concern. We know that this is the limitation of our study that has been stated at the Limitation Paragraph of our revised manuscript.
Point 2: The authors should describe the transfection of T24 cells with PrPC in detail. Was the transfection transient or stable, what expression vector was used in the experiments? Did they use a viral vector?
Response 2: Yes, according to your recommendation, we have provided this information in the Methodology paragraph of our revised manuscript.
Point 3: Line 208. As far as I know, Duke's classification is not used in bladder cancer. It is used in colorectal cancer.
Response 3: We apology for this mistake. We have corrected this inappropriate words in the Result Section (i.e., Figure 1) of our revised manuscript.
Point 4: Line 222. “PrPC could play a cardinal role on BC growth and proliferation.” This conclusion cannot be drawn based on immunohistochemical staining done in this study.
Response 4: Yes, according to your comment, this inappropriate sentence has been rewritten in Result Section (Figure 1) of our revised manuscript.
Point 5: It seems that it is incorrect to compare the intensity score of cytoplasmic staining with that of nuclear staining (Figure 1 G). Instead, the authors should compare the number of tumors with cytoplasmic staining and with nuclear staining.
Response 5: In Figure 1F, the comparison of staining intensity core was in the same patient (i.e., D9 vs. H9), i.e., H9 was the normal section, whereas the D9 was the tumor section. Although it has some positive signal in H9 that maybe due to the fact that harvested sample is not completely appropriate. We notice that the distribution of PrPc signal was predominantly located in the cell nuclei. When we assessed the intensity expression of PrPc, we noticed that PrPc which distributed inside the nuclei had a notably stronger signal than that PrPc located in the cytosol (Fig 1G). This was for why we performed such a calculation. However, we really have no idea for why the PrPc occurred nuclear translocation and what is meaning of this translocation.
Point 6: The authors conclude that normal tissues had lower PrPC expression levels than bladder cancer tissues. However, in figure 1 F, normal urothelium is highly positive for PrPC. Nevertheless, the authors chose interstitial tissue to demonstrate low PrPC expression in this sample. Expression levels and staining patterns (nuclear or cytoplasmic) should be compared between bladder cancer tissues and normal urothelium and not other bladder wall layers.
Response point 6: Dear reviewer, we would like to tell you that initially when we carefully analyzed the immunohistochemical results of cellular prion protein (PrPC) on bladder cancer (BC), we found that the more advanced stage of the BC, the more intensity of PrPC in the nuclei rather than in cytoplasm, implicating that PrPC would translate from the cytoplasm into the nuclei that was really an interesting finding. This could be an important finding that will be further investigated by our Lab in the near future. This means that the strongly expressed PrPc specimen had notably higher nuclear translocation than that of the weakly expressed PrPc specimen in tumor parts. This was for why we had provided such an analytical result in Fig. 1-G.
Point 7: Figure 2 A, B, and C could be arranged in a single line graph with time on the X-axis and cell viability on the Y-axis.
Response 7: Yes, according to your recommendation, we have re-arranged the Figure 2 A, B, and C in a single line graph with time on the X-axis and cell viability on the Y-axis in our revised Figure 2.
Point 8: To further clarify PrPC function in bladder cancer cells, genetic depletion of PrPC using RNA interference should be used.
Response 8: Dear reviewer, first of all, we would like to thank you for your distinctively professional comments. However, we are honorable to tell that three reasons could explain for why we did not perform the study of silencing (siRNA) PrPC gene in the present study:
- Our previous study has conducted such a similar study (Biomedicines 2022 Jan 13;10(1):167):
- A) Protein expression of PrPC, * vs. other groups with different symbols (†, ‡, ), p<0.0001. B) Protein expression of phosphorylated (p)-PI3K, * vs. other groups with different symbols (†, ‡, §), p<0.0001. C) Protein expression of p-Akt, * vs. other groups with different symbols (†, ‡, §), p<0.0001. D) Protein expression of p-m-TOR), * vs. other groups with different symbols (†, ‡, §), p<0.0001. All statistical analyses were performed by one-way ANOVA, followed by Bonferroni multiple comparison post hoc test (n=3 for each group). Symbols (*, †, ‡, §) indicate significance (at 0.05 level). Ct = control group (i.e., H9C2 cells only); ovE-PrPC = overexpression of PrPC in H9C2 cells; siRNA-PrPC = siRNA knockdown PrPC in H9C2 cells; PI3K = Phosphoinositide 3-kinase; m-TOR = mammalian target of rapamycin(mTOR); RAC(Rho family)-alpha serine/threonine-protein kinase.
Therefore, in the present study, we did not want to duplicate the previous study.
- The IJMS Editorial Office requested that we had to resubmit the revised manuscript within 10 days. It is impossible for us to comply with your request.
- We are also honorable to tell you that the financial support for this study had been depleted, implicating that we have no new financial support to finish the additional new data.
We would like to take this opportunity to express our appreciation for your detailed review of the article and the kindness of giving us valuable suggestions. Thank you very, very much!

Round 2
Reviewer 2 Report
The manuscript is drastically improved and is acceptable for publication in its present form.